# IoT-Based Access Management Supported by AI and Blockchains †

**Eryk Schiller** *,‡ 🆔, **Elfat Esati** ‡ **and Burkhard Stiller** ‡ 🆔

Communication Systems Group CSG, Department of Informatics IfI, University of Zürich UZH,
Binzmühlestrasse 14, CH-8050 Zurich, Switzerland

\* Correspondence: schiller@ifi.uzh.ch; Tel.: +41-44-635-4337
† This paper is an extended version of our paper published in CNSM'21.
‡ These authors contributed equally to this work.

**Abstract:** Internet-of-Things (IoT), Artificial Intelligence (AI), and Blockchains (BCs) are essential techniques that are heavily researched and investigated today. This work here specifies, implements, and evaluates an IoT architecture with integrated BC and AI functionality to manage access control based on facial detection and recognition by incorporating the most recent state-of-the-art techniques. The system developed uses IoT devices for video surveillance, AI for face recognition, and BCs for immutable permanent storage to provide excellent properties in terms of image quality, end-to-end delay, and energy efficiency.

**Keywords:** management of access control; video surveillance; blockchains; Internet-of-Things IoT; Artificial Intelligence

## 1. Introduction

Internet-of-Things (IoT) is transforming the world of things, impacting many economic sectors, such as manufacturing, transportation, automotive, consumer goods, and health-care [1]. Thanks to the advances in integrated circuit design, IoT devices are now equipped with powerful processors of a new generation, handling processing loads efficiently [2,3]. This offers an opportunity to run complex tasks on IoT devices in a distributed fashion. However, IoT comes with many challenges or gaps that still need to be improved [4], such as the centralization of various IoT platforms, e.g., Amazon Web Services (AWS)-IoT, security and privacy issues concerning communication protocols as well as vulnerability to various attacks related to the poor maintenance of IoT infrastructures, e.g., Mirai [5].

Blockchains (BC) [6,7] offer immutable storage of data records in a distributed ledger by cryptographic measures. BCs can help IoT infrastructures deal with centralization: when IoT infrastructures store and process data in a BC; this removes the single-point-of-failure present in currently available IoT platforms, such as AWS IoT [4,8–10]. BCs bring significant advantages in terms of information provenance, non-repudiation, and authenticity (while every originator signs every record with its private key), thus increasing the overall information security in the system [11]. Finally, Artificial Intelligence (AI) plays a significant role in providing accurate data analysis in real-time. Nevertheless, the design and development of an efficient data analysis tool using AI come with challenges, such as centralization and transparency [12]. Therefore, integrating BCs with AI can produce a robust approach to resolve those issues. AI is often considered a black box, providing classifiers or predictors, lacking transparency. However, the transparency can be materialized by ordering AI decisions among many nodes in a given BC. This provides a precise, immutable track of AI decisions ordered in time, which can, e.g., form the basis for managerial access control decisions. Therefore, the simultaneous application of IoT, BCs, and AI shows a successful synergy transforming data acquisition, analysis, and storage [11,13,14].

Having a closer look at the research in these three domains mentioned above, one may expect many proposals and architectures. However, no use case exists so far in which those three technologies may complement each other. Therefore, this paper's primary goal is to design and develop a BC-enabled IoT Architecture coupled with AI to efficiently implement a use case, i.e., an access management approach.

The paper's contributions are the following. First, we specify a use case in access management, in which a user is provided with access to a physical resource based on images collected by an IoT device. Second, we implement a real system in which an ESP32-based device [2] takes images and locally runs a Convolutional Neural Network (CNN) of a small size directly on the Micro Controller Unit (MCU) for face detection and face recognition [15]. The implemented infrastructure transports the image from the IoT device to a gateway. The gateway embeds the image and its metadata (i.e., user face identifier) into a transaction and sends the data toward an implemented smart contract. The smart contract runs on top of the Hyper Ledger Fabric (HLF) [7], handles the data received, and stores the data directly within the BC. Therefore, external storage systems such as InterPlanetary File System (https://docs.ipfs.io/ accessed on 25 January 2022) are not needed to handle images. The experimental validation of the system is based on the prototype developed.

Finally, this paper provides a real experience of the system, in which the AI algorithms run on the IoT device, i.e., MCU, directly. The information derived by the IoT-based AI is then provided towards an appropriately selected BC, which can be used as a communication backend or a storage system for auditing purposes.

The remainder of this paper is organized as follows. Section 2 introduces related work. While Section 3 provides a new use case and specifies the system architecture, Section 4 details its implementation and evaluates the performance of the system developed. Finally, Section 5 summarizes the work and adds an outlook on the future work.

## 2. Related Work

The related work addresses recent research on the integration of AI, BC, and IoT.

### 2.1. Internet-of-Things IoT

IoT is transforming our interaction with everyday things [1]. Practically almost anything can be equipped with a microcontroller, sensors, and actuators, thus allowing things to monitor the surrounding environment at a certain level of intelligence. A thing able to react to changing conditions of the environment is referred to as a smart object, which is often also connected to the Internet, allowing for the harvesting of information by centralized storage (e.g., AWS IoT) [4] and more sophisticated information processing at the fog, edge, or cloud level as well as actuating in a given environment.

Most IoT devices depend on microcontrollers [2,3] or microprocessors of minimal processing power, making them much less power-hungry than, e.g., a smartphone. This design choice is based on the assumption that processing power comes with significant energy expenses. Therefore, less efficient computing is energy-efficient, runs with minimal power, conserves energy, and allows the longer lifetime of an object on a single battery charge.

### 2.2. Internet-of-Things and Artificial Intelligence

Due to the low processing power of IoT devices, there is a need for green communication in the IoT domain [16] as well as lightweight approaches in AI. As a result, different algorithmic approaches are taking place, and more efficient algorithms are being developed [17]. This gives rise to novel research domains, such as Tiny Machine Learning [18] and Tiny Deep Learning [19], which can operate on constrained IoT devices. As an example, MIT researchers [20] have implemented a system called MCUNet, which has a high potential to bring deep learning to low-capacity devices such as tiny microcontrollers for performing tasks such as image, audio, or video recognition.

One of the most recent studies [17] worked in the direction of image processing and cloud offloading. Two approaches with deep learning were tested, i.e., (*i*) cloud offloading

of deep learning platforms and (*ii*) migrating deep learning to IoT devices. The two approaches were tested and looked at from the perspective of reduced energy efficiency and real-time requirements of object recognition. In the first approach, they used CNNs on the cloud, while the device was responsible for taking images and forwarding them to the cloud. The results show that executing machine learning on an IoT device consumes more energy than cloud offloading. However, AI cloud offloading also has drawbacks, leading to a latency starting from 2 s that goes up to 5 s, which might be longer than the execution of small AI on IoT devices. This infers the response time variability, making it unreliable and not valid for real-time AI image processing.

Esp Eye [2], equipped with Tensilica LX6 dual-core processor, is most likely the first microcontroller device that performs real-time face recognition. However, this does not mean regular face detection and recognition algorithms have been used here. Esp Eye is one of its kind that comes with the Esp Who [15] platform, which supports both face detection and face recognition. To our knowledge, this is the only device that can perform video streaming combined with real-time face recognition in a microprocessor that lies outside the main three classes of Central Processing Units (CPUs) such as Intel, AMD, or ARM processors. Esp Who platform implements a framework called MTMN. MTMN refers to both Multi-Task Cascaded Convolutional Networks (MTCNN) [21] and MobileNets [22] as well as Face Recognition model based on Convolutional Neural Network (FRMN) [23]. Several deep learning techniques have been specified and implemented that paved the way towards face detection. However, MTCNN is a framework that integrates both face detection and alignment. With the help of MobileNets, it builds lightweight deep neural networks, which use depth-wise separable convolutions for face detection. The FRMN is a Convolutional Neural Network mixing local and global image features. It provides good feature extraction from the mouth, nose, and eyes, delivering enhanced and accurate face recognition.

### 2.2.1. Convolutional Neural Networks

CNN emerged with the development of LeNet [24,25]. LeNet was limited to hand-written digit identification, which could not scale efficiently to all image classes.

### 2.2.2. Multi-Task Cascaded Convolutional Networks

MTCNN is one of the state-of-the-art approaches [21] for detecting faces in images. MTCNN can achieve 95% accuracy on a range of benchmark datasets. MTCNN performs a CNN-based framework that simultaneously performs face detection and alignment, while other CNN-based frameworks treat face detection and alignment as two distinct processes.

First, once the image is captured, it is then scaled into multiple different sizes based on different scaling ratios, forming a collection of images called the Image Pyramid [26] that allows the processing of images at different scales simultaneously. This image processing is used because it makes detecting faces easier, no matter how far or close they stand in the image.

The process consists of three CNN stages for every image in the Image Pyramid, i.e., P-Net, R-Net, and O-Net, capable of detecting faces and landmark locations such as eyes, nose, and mouth. The three mentioned stages perform independently of each other, while the output one is used as the input of another. Every step applies Non-Maximum Suppression (NMS) to merge highly overlapped image candidates. A short description of the MTCNN stages is provided below.

#### Proposal Network (P-Net)

This CNN (cf. Figure 1a) is an initial processing stage that obtains images at different scales from the Image Pyramid. P-Net places a rectangle over a detected face and provides landmark candidate positions.

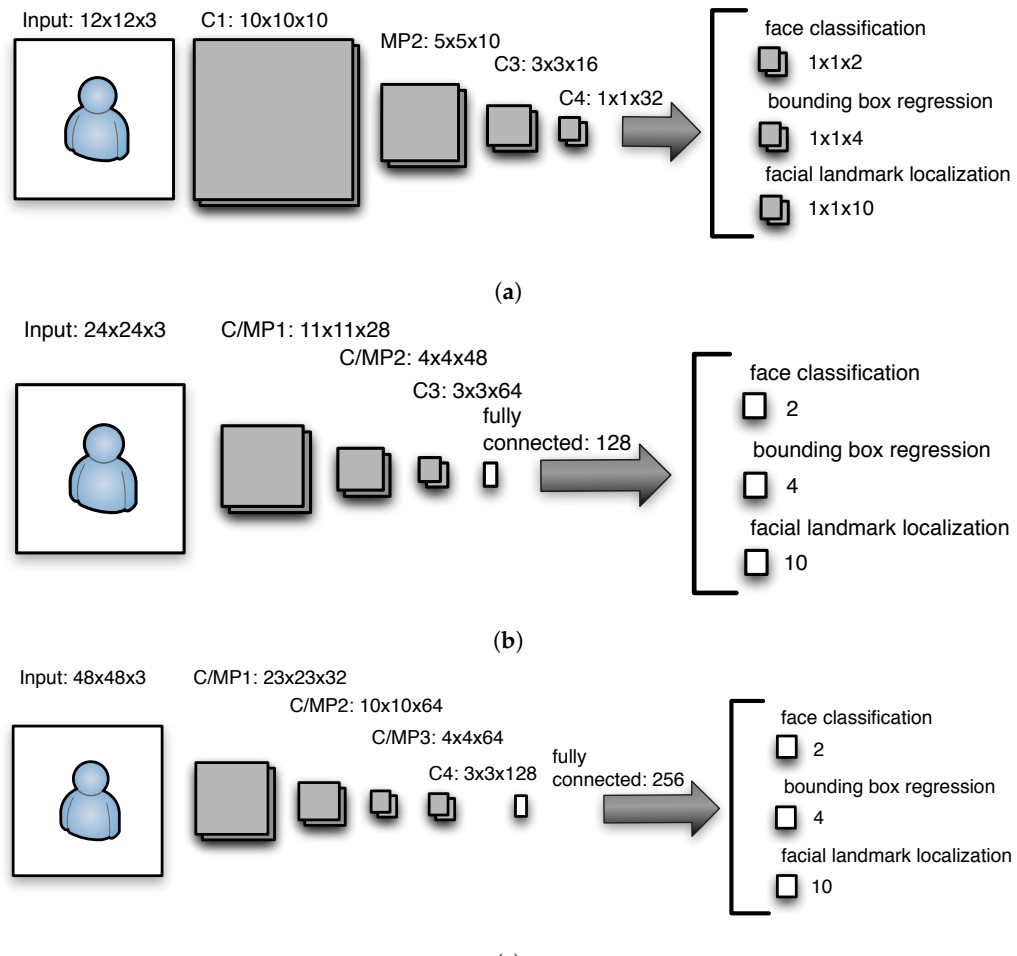

**Figure 1.** MTCNN CNNs (**a**) P-Net, (**b**) R-Net, (**c**) O-Net.

Refine Network (R-Net)

The structure of R-Net is displayed in Figure 1c. The input of this stage is the bounding box generated by P-Net. R-Net filters out boxes with low confidence by using a higher input resolution. This stage helps to alleviate P-Net false-positives.

Output Network or (O-Net)

O-Net (cf. Figure 1c) is the final processing stage of MTCNN, where increased resolution is added and accuracy is significantly increased. In this case, the output of R-Net serves as the input of this stage.

2.2.3. Face Recognition Model Based on CNN

MobileFaceNets [22] consists of MobileNetsV2 [27] and ArcFace algorithm. The output of face detection is an aligned face with landmark features. It becomes the input for face recognition. If a human is detected using the aforementioned model, the face recognition algorithm executes to generate a face identifier (ID). In order to verify a person, MobileFaceNets has to check if that person exists in a face database. It relies on comparing the newly generated face ID with Face IDs already existing in the database. Most likely, there may not be an exact match for face IDs, even if it is from the same person. Hence the idea is to obtain the distance between the face IDs, which is done by Euclidean or arc distance. In order to determine if the face ID is from the same person, MobileFaceNets compares the distance between the Face IDs based on the allowed threshold. The MobileFaceNets model utilizes deep CNNs, i.e., standardized CNNs of many layers deep, to output a discriminative vector of 512 values.

### 2.2.4. MobileNets for Lightweight CNNs

MobileNet (MN) [22] is a CNN architecture developed by Google, which builds lightweight deep neural networks and uses less computational power than regular CNNs. This architecture allows face detection and recognition to be run on IoT devices, mobile devices, and computers with low computational efficiency without compromising the accuracy of the results. MobilNetV2 [28] is a recent version that comes with refinements of the MobileNet and the introduction of residual blocks, with skip connections to connect the beginning and finish of a convolutional block. By combining these states, the network can now retrieve prior activations that were not updated in the convolutional block, which proves essential in deep networks. The main idea behind MobileNetV2 is to obtain high accuracy with less computational power.

#### Regular Convolution

In a regular convolution, the convolution kernel or the filter is applied simultaneously to all channels of the input tensor. Let us assume an input tensor $D_F \times D_F \times M$, i.e., $D_F \times D_F$ of $M$ channels, convolution stride 1, a convolution kernel of size $D_K \times D_K$, and an output tensor $(D_F - D_K + 1) \times (D_F - D_K + 1) \times N$. In such a case, the kernel is of size of $D_K \times D_K \times M \times N$, and the convolution has a computing effort of $D_K \times D_K \times M \times N \times D_F \times D_F$ expressed in Multiply Additions (MAdds).

MobileNets has a different approach. It employees the depthwise convolution using two parts, i.e., depthwise convolution and pointwise convolution.

#### Depthwise Convolution

With depthwise convolution instead of applying one kernel to $M$ channels simultaneously, MobileNets apply a kernel of size $D_K \times D_K \times 1$ to every channel of the input tensor separately, i.e., $D_F \times D_F \times 1$. In such a case, there are $M$ kernels of size $D_K \times D_K$, which results in $D_K \times D_K \times M \times D_F \times D_F$ operations, which transform an input tensor into a $(D_F - D_K + 1) \times (D_F - D_K + 1) \times M$ output tensor.

#### Pointwise Convolution

To increase the number of channels in the output tensor, a pointwise convolution is used. Pointwise convolution applies a kernel of size $1 \times 1 \times M$. To obtain $N$ channels, pointwise convolution applies the same $1 \times 1 \times M$ kernel $N$ times. In this way, MobileNet obtains the same size of feature map as in regular convolutions but with a lower number of computations.

#### Parameters and Computing Effort

The number of parameters of a regular and MobileNets convolution is calculated as $D_K \times D_K \times M \times N$ and $D_K \times D_K \times M + M$ respectively, while the number of operations for a regular and MobileNet convolution is calculated as $D_K \times D_K \times M \times N \times D_F \times D_F$ and $D_K \times D_K \times M \times D_F \times D_F + N \times M \times D_F \times D_F$. Thus, MobileNets are enablers of lightweight computing in IoT architectures, while the number of parameters and the number of operations decreases by a large factor. As an example, the number of operations with the help MobileNets decreases by a factor of $1/N + 1/D_K^2$ in comparison to regular convolutions (similar calculations can be consulted using the following link https://towardsdatascience.com/review-mobilenetv1-depthwise-separable-convolution-light-weight-model-a382df364b69 accessed on 25 January 2022).

### 2.3. Internet-of-Things and Blockchains

Narrowing down the scope to BC and IoT systems, one sees research attempts to close the gaps of IoT systems by removing the centralized control as well as attack the problem of provenance, non-repudiation, and authenticity in IoT data streams with the help of BC [4,8–10,29]. Other studies [30] attempt to close security and privacy IoT gaps with the help of the BC to ensure the reliability and availability of the data.

### 2.4. Artificial Intelligence and Blockchains

There is a high research interest in BC and AI analyzed in various domains and applications. Some research [31] focuses on the application of AI in the BC for making BCs more efficient. It concentrates on BC consensus mechanisms and better governance. There are also research items about the applications of BC in AI. Like IoT, the AI domain also suffers from security, centralized architecture, and resource limitations. This is what BC promises to solve. There is a lot of discussion and research in this area. However, most of these are reviews and solutions that do not develop use cases or provide actual implementations.

### 2.5. Artificial Intelligence, Blockchains, and Internet-of-Things

Several researchers have attempted to shed light on the benefits of converging IoT, BC, and AI [11]. However, most of these attempts are either reviews or explorations that lack a concrete implementation in a use case [13,14]. A more comparative research outcome in this domain is a so-called BlockIoTIntelligence attempt [12], which proposes an architecture that utilizes BCs and AI in IoT. BlockIoTIntelligence aims to achieve decentralized big data analysis considering the security and centralization issues of IoT applications in various domains such as smart city, healthcare, and intelligent transportation. BlockIoTIntelligence claims the mitigation of existing challenges to obtain high accuracy, reasonable latency, and security. Another work examined safe data sharing in Mobile Edge Computing (MEC) systems supported by blockchain technology. An adaptively privacy-preserving technique was suggested to safeguard users' privacy in data sharing. Additionally, the energy consumption of the MEC system and the blockchain transaction performance was jointly improved through an asynchronous learning approach. The performance of this work was studied through simulations [32].

### 2.6. Approaches Similar to This Work

A use case similar to this work [33] is the design and implementation of a camera-based sensor for room capacity monitoring. That work aims to count the number of people present in a room with the help of a Raspberry Pi (RPI) [3] device equipped with a camera. Their architecture employs AI and IoT. The role of the camera is to take pictures. The pictures were analyzed with Machine Learning (ML). When the analysis is complete, the data are sent to a LoraWAN Server [34]. To this end, they have attached a LoraWAN modem to the RPI. Furthermore, a web application shows the occupation of a given room. In that work, face detection is performed directly on the RPI. Finally, the algorithm counts the number of people entering the room and reports the number to the LoraWAN back-end server. That work still depends on a central server. Furthermore, the data are stored in a database. Finally, security issues are not considered. This approach differs from our work because the BC part is absent in [33]. Furthermore, a camera is attached to a powerful RPI node equipped with a regular Linux operating system.

### 2.7. The Newly Proposed Approach

This approach differs from related work implementations by providing a solid use case in access management. Furthermore, this approach combines all three techniques at the same time, i.e., AI for image processing, BC for immutable tamper-resistant storage (e.g., for auditing reasons), and IoT for data harvesting (i.e., providing the video stream). Furthermore, this approach follows the novel TinyML paradigm, in which face detection and recognition run directly on an IoT device.

## 3. Use Case and Architecture

Driven by the specific use case, this paper provides an extensive description of the architecture and gathers an in-depth analysis of the system previously briefly presented and superficially evaluated in a short conference proceedings paper [35].

### 3.1. Use Case

The system employees real-time face detection and recognition of authorized individuals to grant access to an institution. The access is granted or denied by the system automatically. For example, when access is granted, the door to an institution may open automatically without any intervention. However, the door will remain closed when access is denied, preventing the user from accessing a given resource. Every time an individual needs access, their picture is taken, processed, and stored in the immutable BC, preventing future tampering with data and enabling immutable storage that provides a solid foundation for auditing purposes.

This use case (cf. Figure 2) assumes that access is revoked when an individual is not registered in the system. However, a picture of unknown individuals is still taken and stored in the BC for auditing purposes. Once a stranger is detected continuously, a warning is raised to inform the administrator about this issue. Simultaneously, the number of people entering and leaving an institution or a room can be tracked. Tracking the number of people entering or leaving the room can be very beneficial in many situations, such as capacity tracking, infection spread.

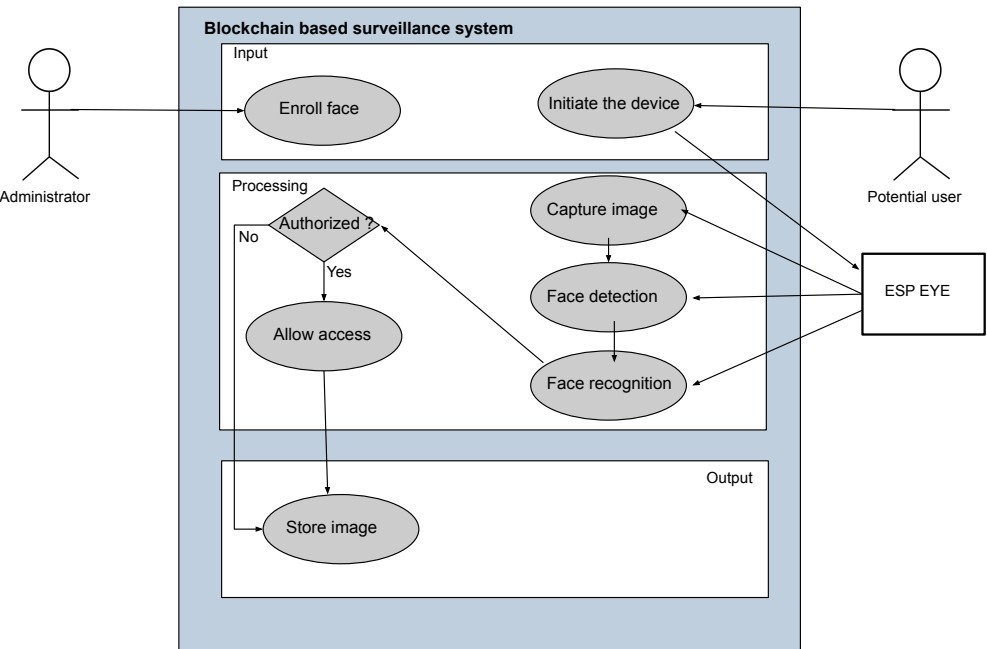

**Figure 2.** High-Level Use Case Overview.

The process involves an individual approaching an entry point of an institution or a room, where an IoT device equipped with a camera detects and recognizes the person and grants or rejects access. Face recognition is performed with the help of face alignment, detection, and recognition. The images processed are forwarded to the BC, where they are stored in an immutable data structure. The system raises alarms if a continuous presence in front of the device is detected, where a person may want to perform a security breach.

### 3.2. Architecture

Based on the hardware components available, the software architecture design and the reasons for these design decisions taken to materialize the idea of IoT-based AI surveillance with BC are provided. Furthermore, different approaches, technologies, and communication protocols, considered throughout the design decisions, are described.

### 3.2.1. Hardware Components

The image capturing and face recognition are handled by Esp Eye [2] directly, whereas the BC environment is based upon an Intel-based machine running a macOS. Figure 3, top left-hand side, displays multiple Esp Eye devices, which communicate with the IoT gateway within the same WiFi-based (i.e., IEEE 802.11) network. The IoT Gateway serves as the middle man, which waits for data (i.e., images and metadata) coming from Esp Eye devices to be inserted into the BC. The communication between Esp Eye devices and the IoT Gateway is achieved through a wireless communication compliant to the IEEE 802.11 Local Area Network (LAN) protocol. To provide integrated experimental facilities, the decision was to run a BC locally within the platform provided. However, the BC can be spanned among multiple machines organized as a BC network on the Internet without any hassle. Since HLF [7] was selected as the BC platform and its official build is provided for Intel-based CPUs, the decision was to run the IoT gateway on an Intel-based machine. It is, however, expected that HLF may run on low-capacity devices, such as ARM-based RPI devices. To this end, the HLF developer (i.e., IBM) shall provide an appropriate compilation environment to support ARM-based devices as well.

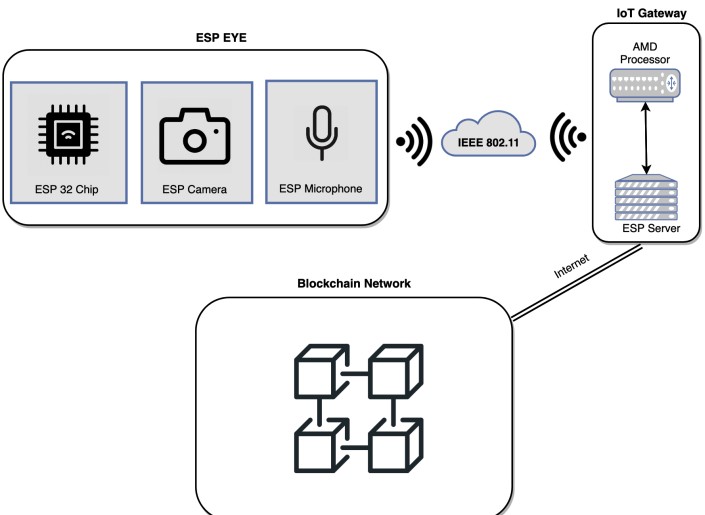

**Figure 3.** Hardware Architecture.

Esp Eye [2] integrates an embedded microphone and a camera with two megapixels. The camera is an OV2640 sensor with a maximum image size of 1600 × 1200 pixels. The board supports 2.4 GHz WiFi technology to connect to the Internet through a Local Area Network (LAN). A Micro USB port provides the power supply and also allows for debugging. In addition, it comes with a Universal Asynchronous Receiver-Transmitter (UART) port, which enables asynchronous serial communication to program the Esp Eye device. Esp Eye also supports several security features, such as flash encryption and secure boot. The flash encryption is intended to secure the content of the flash memory. In Esp Eye, flash encryption is performed using AES-256, and the key is stored in the eFuse (i.e., special-purpose storage on the chip). eFuse keeps the values intact and cannot be changed by a software. Furthermore, since the data on the flash are encrypted, a physical readout will not be possible. Additionally, secure boot can protect the device from uploading unsigned code. The device uses a typical digital signature method with the Rivest–Shamir–Adleman (RSA) cryptography. The public key is stored in the device itself, whereas the private key is kept secret and used upon each code upload. Apart from security, Esp Eye is an outstanding device due to its performance (i.e., a double-core architecture supporting the 240 MHz CPU frequency) and a face detection/recognition platform known as Esp Who. As elaborated in Section 2.2, Esp Who comes with algorithms for face detection (i.e., MTMN) and recognition (i.e., FRMN), which both run directly on the Esp Eye device.

### 3.2.2. Software Architecture

Figure 4 depicts a high-level overview of the software components. It is essential to mention that the software design shall be compatible with many underlying hardware architectures. However, the Esp Eye is required for the success of this project.

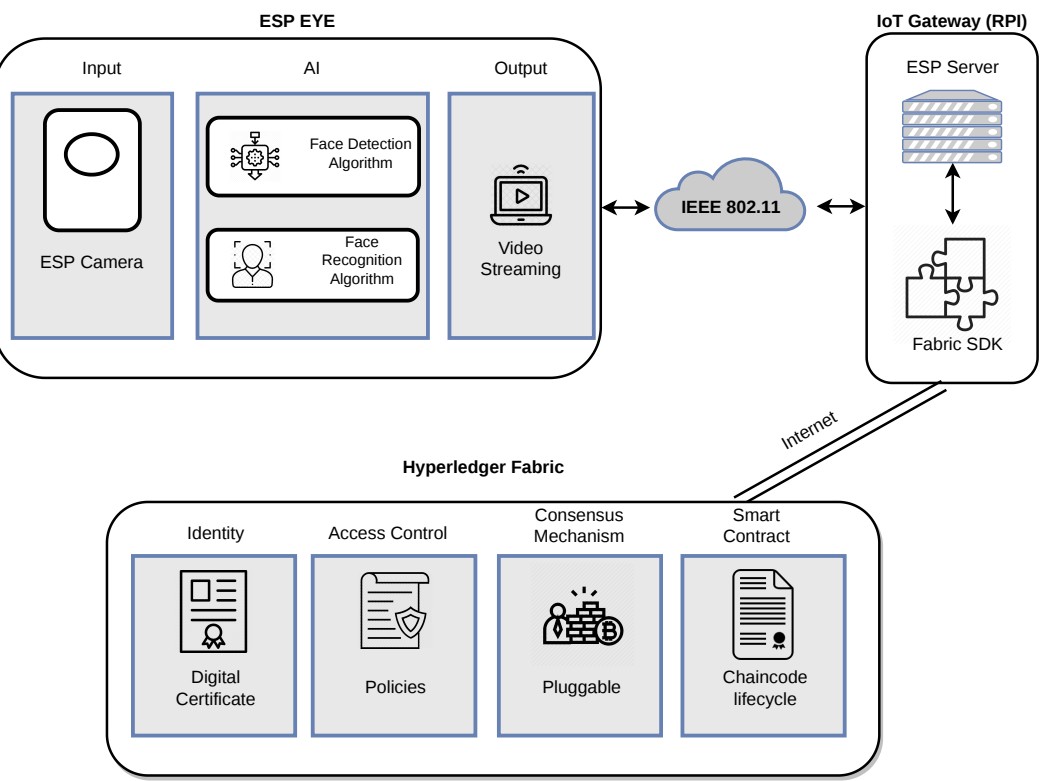

**Figure 4.** Software Architecture.

At the beginning, two approaches were considered: deployment of the face detection/recognition on the cloud and on the Esp Eye itself. In the first approach, the image processing AI is deployed on the cloud, which means the camera of a sensor captures images and sends them toward the cloud for face recognition. However, this centralized approach suffered from a single-point-of-failure and was later discontinued. In this work, Esp Eye uses the camera to capture images. Images are sent to the face detection and face recognition modules. It is noteworthy that face detection (i.e., MTMN [21,22]) and face recognition (i.e., FRMN [23]) run directly on the Esp Eye device. The outcome, i.e., an image accompanied by meta-data, is provided toward the implemented video streaming service, which connects with the implemented Esp Server running on the IoT Gateway. The Esp Server provides the image toward the implemented ESP Plugin (cf. the IoT Gateway), with the help of the HLF Software Development Kit (SDK), to submit the Transaction (TX) to the immutable ledger. HLF stores images coming from Esp Eye devices in immutable storage. HLF runs specific services required to run a BC. It consists of identity management, access control, and a consensus mechanism. Finally, HLF maintains the heart of the system, which is the smart contract implemented in this work that is responsible for handling data received from its clients (i.e., Esp Plugin residing on the IoT Gateway). HLF matches very well the use case (cf. Section 3.1) because a private permissioned BC is better suited for video surveillance due to privacy, BC block size, and BC block time reasons. The data gathered by a given organization have to remain protected against malicious third parties. This work also considered other BCs, for example, Ethereum (ETH) [6]. Although ETH is a public permissionless BC, it can still be adapted to the needs of this work (e.g., by running the private network or encrypting data on the chain). However, ETH comes with a Proof-of-Work (PoW) consensus mechanism, which is energy-demanding and costly. In

contrast, HLF uses a modular architecture; it enables flexibility in selecting pluggable consensus mechanisms from a broader spectrum of candidate algorithms. Furthermore, HLF is fast, while some configurations allow for high workloads exceeding 20,000 supported TXs per second [36]. Additionally, HLF does not involve additional costs in terms of TX execution. Moreover, Hyperledger Fabric allows for splitting the network into multiple smaller networks (i.e., channels) to separate competing tenants and further improve privacy. Finally, HLF supports arbitrary user data of large sizes in TXs that do not hold for other BCs, which allow only short user-generated messages [37]. In our previous work [8,10], BAZO BC was used, which did not allow the storage of large portions of user data, such as images. We, therefore, decided to move on to a more suitable BC, i.e., HLF, rather than using less flexible BCs and storing data (i.e., images) off the chain.

### 3.2.3. Communication Protocols

Considering communication protocols, Esp Eye is attached to a WiFi network, which provides Internet Protocol (IP) communication. This work considered two major protocols, i.e., Message Queuing Telemetry Transport (MQTT) and Hypertext Transfer Protocol (HTTP), for the actual data transmission on the application layer, both supported by the Esp Eye device.

MQTT is a push-based client-server protocol that follows the Publish/Subscribe (Pub/Sub) communication model. Furthermore, MQTT is data-centric. Since video surveillance works with images, the decision was made to move towards document-oriented communication instead.

HTTP is a document-oriented client-server communication protocol. HTTP is employed between Esp Eye and the IoT Gateways to communicate in the client–server architecture. The Representational State Transfer (REST) architectural style is used for inter-machine communication because REST is considered a lightweight communication paradigm. The JavaScript Object Notation (JSON) data-interchange format [38] is employed to carry the actual information in the system. JSON is a lightweight format of syntax following the Javascript language notation. It makes JSON easy to read and process with high-level languages such as Javascript (JS).

### 3.3. Implementation

The implementation encompasses the data flow between Esp Eye, IoT Gateway, and HLF as shown in Figure 5, where data circulates from the left (i.e., the Esp Eye device) to the right (i.e., HLF) [39]. Several Application Programming Interfaces (API) and data structures are used to materialize the system.

Esp Who [15] analyses each frame with the MTMN face detection and if a face is detected, the image is provided towards the FRMN face recognition algorithm.

### 3.3.1. Multi-Task Cascaded Convolutional Networks (MTCNN) with MobileNets (MN)

MTMN is an already trained model for face detection, which is based on the MTCNN implementation using the MobileNetsV2 convolution framework that significantly reduces the number of operations expressed in terms of MAdds and the number of model parameters. Table 1 gathers the number of parameters and number of operations in convolutions for each processing step of the MTCNN convolution network using regular convolutions (MTCNN) and MobileNetsV2. The calculations are performed for all convolutions within a given MTCNN step according to Section 2.2.4 (cf. Figure 1).

It can be observed that the MTMN model size, when MobileNetsV2 convolutions are used, is very compact. Complexity-wise, it is worth noting that MTCNN/MTMN executes more complex stages (e.g., R-Net) only in the case of the previous operation (e.g., P-Net) suspecting a face in a given area of the picture. Thus, heavy processing is performed for picture areas with suspected face candidates. As input, MTMN acquires all images from the image pyramid constructed from the picture scaled accordingly.

**Table 1.** MTCNN vs. MTMN.

| CNN Stage | Regular | | MobileNet | |
|---|---|---|---|---|
| | Parameters | Operations (MAdds) | Parameters | Operations (MAdds) |
| P-Net | 6318 | 116,352 | 327 | 20,362 |
| R-Net | 25,139 | 2,095,680 | 550 | 309,276 |
| O-Net | 135,168 | 15,951,872 | 1310 | 2,121,504 |

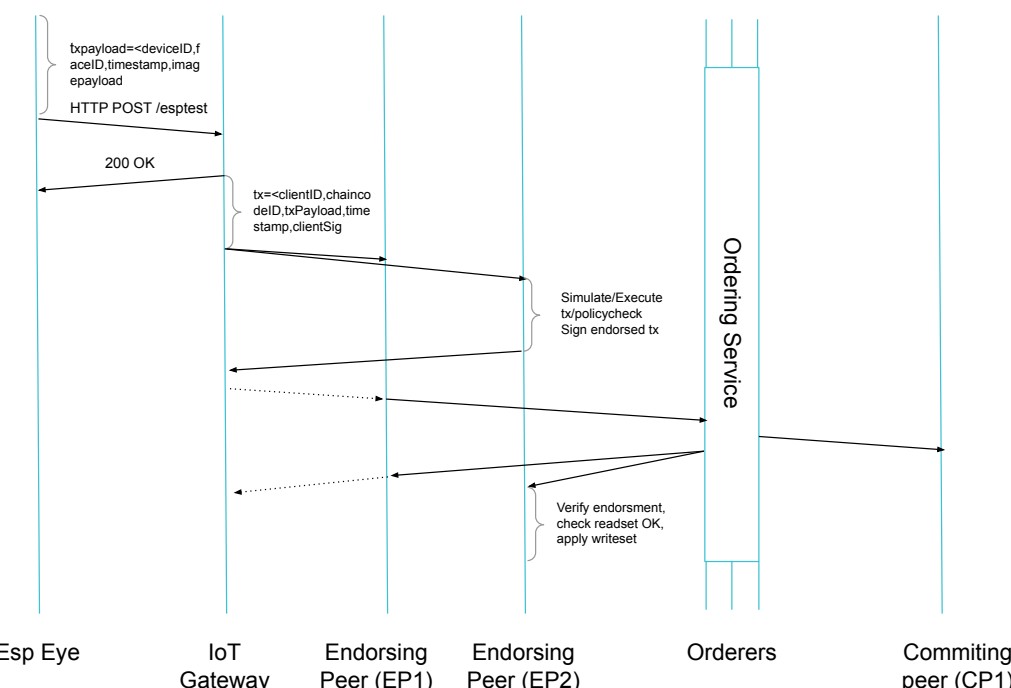

**Figure 5.** Sequence Diagram Showing TX Execution.

### 3.3.2. Face Recognition MobileNets (FRMN)

FRMN is also part of the Esp Who [15] framework. By default, FRMN uses an adapted MobileNetV2 network, which is tuned to receive $56 \times 56 \times 3$ images, and provide the output vector of 512 parameters (2 KB). FRMN can use the Euclidean or arc distances to compute distances between faces. The MobileNet used is a Deep Neural Network (DNN) with 0.5 mln parameters (2 MB) and has an execution time of 56 M MAdds. The size of the model provided by Esp Who is massive and takes a large portion of the available memory on the Esp Eye chip.

### 3.3.3. Note on MTMN and FRMN Complexity

It is, therefore, concluded that CNN is a well-suited method for IoT devices in terms of face detection and face recognition. However, CNN has to be used in conjunction with techniques optimizing the number of CNN parameters (i.e., RAM) and operations (i.e., CPU) on constrained devices. MTMN and FRMN from Esp Who [15] are lightweight versions of more heavyweight models provided in the literature [21,22], where FRMN is much more expensive than MTMN both in the case of RAM and CPU usage.

### 3.3.4. Esp Eye Transmission Overview

Hence, if a face is detected, but not necessarily recognized, the HTTP client is activated, while Esp Eye devices act as clients communicating with the remote server on the IoT Gateway. The video service takes the image equipped with meta-data (i.e., device ID, detected face ID, timestamp) and forwards it to the IoT Gateway. This is achieved with the help of a REST API call using the HTTP POST request. The node.js [40]-based server

located in the IoT Gateway receives the request (e.g., an image with the details of a person detected provided as meta-data). Furthermore, the role of the node.js server is to properly acknowledge the successful reception of the transmission coming from Esp Eye devices.

There are four significant parameters to be stored in the BC reflected in the JSON document (cf. Figure 6a) provided by Esp Eye devices. First, as multiple Esp Eye devices may be employed in access management, the device ID is essential, since the framework needs to distinguish particular devices from which the information is coming. Second, a face ID is needed, allowing for personal identification without processing the captured frame again. The successfully identified person on the sensor implies that access was granted to given resources protected by this access management system. Additionally, the timestamp identifies the time moment when the person is detected. Finally, the image frame is encoded in BASE64 [41]. As shown in Figure 5, all four parameters are sent as a JSON document. After receiving the document, the node.js server located on the IoT Gateway responds with a status code. The Esp Eye is programmed with the help of the Arduino IDE [42], which has to be equipped with the ESP32 board support. Consequently, the appropriate board called ESP-CAM has to be selected to program the Esp Eye device. The WiFiClient library (https://www.arduino.cc/en/Reference/WiFiClient accessed on 25 January 2022) enables establishing the connection to the IoT Gateway by using its IP address. The Esp Eye issues HTTP requests using the POST method towards the HTTP API exposed by the IoT Gateway. For submitting JSON documents, the *application/json* method is used, which is a standard format for sending structured data. JSON is handful for sending plain text or any other data types. Since HLF also uses a JSON format to store assets, this work converts the image into a data type supported by JSON as well. Hence, the best option is to store the image as a string. To this end, the image is first converted to BASE64. The Esp Eye converts images to BASE64 and encapsulate the BASE64 representation into a JSON format.

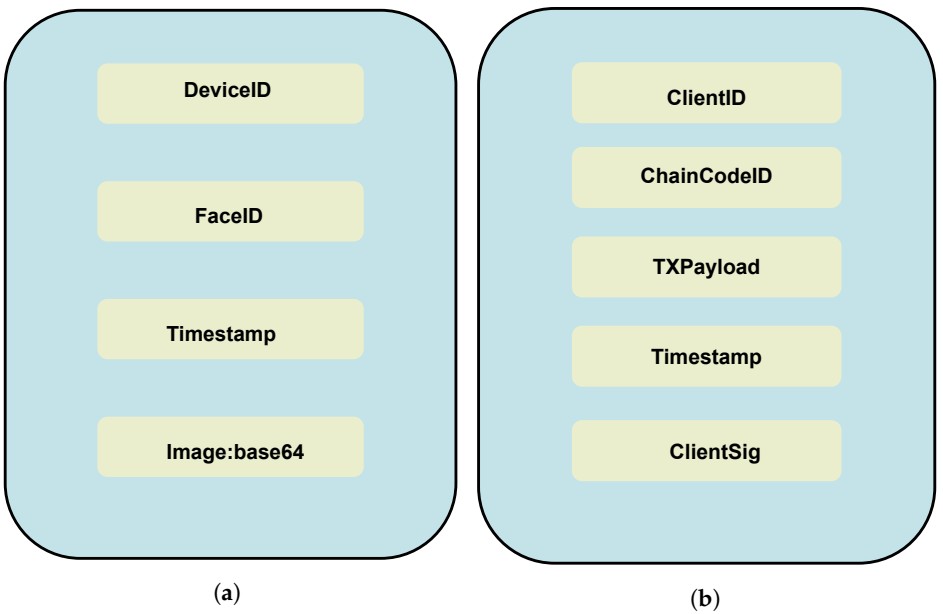

(a)  (b)

**Figure 6.** JSON Formats Used: (**a**) Sent From Esp Eye Devices, (**b**) Included in the HLF TX Proposed.

### 3.3.5. HLF Transaction Submission

The node.js HTTP server is responsible for receiving the data and forwarding it to the HLF. In order to interact with HLF, the node.js-based server uses the HLF SDK providing an API to submit TX to the ledger. The process of submission takes place right after the image has arrived from an Esp Eye device. The HLF TX data, also referred to as an asset, is a collection of key-value pairs. Since the JSON format for the image transfer is used from the start, node.js may forward a similar JSON file to the BC.

Before the HLF TX is initiated, three preconditions have to be fulfilled. (*i*) The HLF has to be configured and running. (*ii*) The HLF channel is established (i.e., HLF subnetwork for privacy concerns). (*iii*) The chaincode implemented in this work (i.e., HLF Smart Contract identified by its chaincode ID) with its endorsement policy is deployed within a given channel. Every Esp Eye is registered and enrolled with Certificate Authority (CA), while the Membership Service Providers (MSP) maintain the identities of every Esp Eye device. For more information, please consult the HLF documentation [7].

The IoT Gateway acts as an HLF client. Therefore, the IoT Gateway initiates the HLF TX with these four parameters provided as a JSON document by Esp Eye (cf. Section 3.3.4), which becomes the TX payload. Each Esp Eye is assigned with a public/private key pair, which is used to sign the HLF TX. In this work, the IoT Gateway retrieves the key pair of a given Esp Eye device and signs the TX on behalf of the device using the HLF SDK. Although TX signing is possible in the offline mode, meaning locally on the Esp Eye device, this was not implemented here, since the HLF client SDK is not ported for Esp Eye devices yet.

Typical BCs, such as ETH, use the order-and-execute paradigm, meaning the TX is first ordered in blocks and later on executed by all peers in a given BC network. The speed of the HLF environment is achieved through the application of a so-called execute-and-order TX submission scheme, which depends on the configurable number of endorsement peers, which first execute all TXs received. Endorsers endorse a given TX. When the HLF TX receives endorsements, which satisfies the configured endorsement policy, the TX can be ordered in blocks by the ordering service according to a configured consensus mechanism and committed into the ledger by committing peers.

This work configures two endorsers (i.e., EP1 and EP2, cf. Figure 5). They receive the TX proposal, which includes a client ID, the chaincode ID (i.e., HLF smart-contract ID), the TX payload, a timestamp, and the clientSignature as presented in Figure 6b. Only endorsing peers specified by the chaincode receive this TX proposal. The endorsement policy in this work requires both endorsement peers to endorse the received HLF TX before the TX might be submitted toward the ordering service. First, EP1 and EP2 check the format of the TX. Second, every TX has to possess a valid signature of a client appropriately registered within the MSP. Third, the client has an authorized member of the HLF channel. When all conditions have been checked, the endorsers invoke the chaincode using the JSON document received. Eventually, the TX is executed; however, it does not yet update the ledger. Now, the endorsers sign the proposed TX and send it back to the HLF client on the IoT Gateway. The intent of the HLF client is to submit the TX to the ordering service and update the ledger. If the HLF client's intention was to query the ledger, there is no need to submit anything to the ordering service.

Before the HLF submits the final version of the TX, HLF clients send the TX endorsed toward the ordering service. Now, the TX is equipped with signatures of endorsing peers. While the ordering service may receive TXs from other clients or ESP devices, the ordering service orders TXs according to a sequence number and packages them into blocks. When the maximum number of TXs allowed in a block is reached or the maximum block-time has passed, blocks are sent to committing peers to be included in the ledger for an immutable storage.

Upon receiving a broadcast message with the created block from the orderers, committing peers verify the signatures of ordering nodes within a given block. HLF allows for configuring committing peers. However, typically all peers in a given channel may update the ledger. If the committing peers fail to verify the signature of ordering peers, the ledger will not include them and rejects the newly created block.

## 4. Evaluation

Important criteria for an evaluation include the reliability of the overall design, end-to-end delay from the moment the image is taken until it is finally submitted into the BC, the quality of images captured by Esp Eye as well as stored in Hyperledger Fabric, and the energy efficiency of the solution from an IoT device's perspective. The evaluation integrates

two Esp Eye sensors (dual-core Tensilica LX6 processor with a maximum frequency of 240 MHz, 4 MB PSRAM, and 4 MB flash) and regular macOS-based computer with 2-core Intel Core i5 running at 2.7 GHz, 512 GB SDD disk, and 8 GB RAM. To begin testing, several steps are recommended. (*i*) The Esp Eye sensors are powered up using an external charger power bank; 10 face profiles are uploaded on the device. (*ii*) HLF is started with the help of docker containers. Currently, one committing peer and two endorsing peers are supported. The ordering service is set to *solo*, i.e., one node submitting HLF TXs. The channel is configured and JS-based chaincode is deployed. (*iii*) The IoT Gateway starts the node.js-based HTTP server listening on port 8585.

### 4.1. Image Quality

The Esp Who platform supports the OV2640 camera embedded in Esp Eye. The camera can be configured in terms of frame size and pixel format. The frame size may be set to one of the following options: FRAMESIZE_CIF (400 × 296), FRAMESIZE_QVGA (320 × 240), FRAMESIZE_VGA (640 × 480), FRAMESIZE_SVGA (800 × 600), FRAME-SIZE_XGA (1024 × 768), FRAMESIZE_SXGA (1280 × 1024), and FRAMESIZE_UXGA (1600 × 1200). With FRAMESIZE_QVGA and FRAMESIZE_CIF, the Esp Eye with face detection and recognition run almost equally well. One difference between them is the number of frames per second delivered. The larger the image size, the more processing time it takes to receive the image. Therefore, due to the higher width, the FRAMESIZE_CIF consumes more processing power, and it can achieve on average 3.2 fps (i.e., 312 ms to deliver an image). In contrast, with FRAMESIZE_QVGA, the sensor can deliver 5.2 fps (i.e., 190 ms to deliver an image). Therefore, FRAMESIZE_QVGA was selected as the image resolution used in this work.

### 4.2. Processing Delay of Face Detection and Recognition

First of all, the idea of performing face detection and recognition on Esp Eye is a novel approach because typically face detection and recognition run either on the cloud or on a local computer. In related work (cf. Section 2), it was previously reported that face detection and recognition offloading to the cloud takes at least 2 s and sometimes goes up to 5 s.

The real experiment was performed, in which the Esp Eye [2] sensor observed its surroundings. Two situations were considered. In the first situation, no face is presented in front of the camera. In the second case, the person registered with the sensor presents his face in front of the camera at a distance of around 0.4 m. The analysis that followed was based on the log of the processing time performed by Esp Eye. The processing was repeated up to five times. For instance, in the situation when no face is detected in the image delivered (Figure 7), the overall processing needs 174 ms to accomplish, which involves 122 ms to receive an image from the sensor, 52 ms to perform the face detection with MTMN, and 0 ms to perform the face recognition with FRMN. Since MTMN has detected no face, FRMN was not activated, resulting in 0 ms completion time. The standard deviation calculated for all processing stages (i.e., Reading the Camera, MTMN, and FRMN) is presented in the figure as well. It is noted that the average overall processing time varies only a little bit with the relative variation of less than 1.9% experienced in all the experiments performed. Face detection and recognition take on average 1008 ms (cf. Figure 8), which includes 126 ms for image acquisition, 172 ms for face detection, and 710 ms for face recognition. The overall processing time experiences a relative variation of around 7.9% in this experiment.

Furthermore, it is noted that MTMN displays accuracy of around 10%, as the face is discovered in 1–2 images for every 10 images taken. This is, however, evaluated positively, as the speed of MTMN is very fast, and several images can be analyzed per second (cf. Figure 8). FRMN displays higher performance as a face is recognized in one of every two images. The detection rate of 50% is understandable when considering the small size of the CNN. Robust networks (e.g., MobileFaceNets [22]) have much higher number of parameters (e.g., 1 M) and require higher computing cost (e.g., 221 M MAdds).

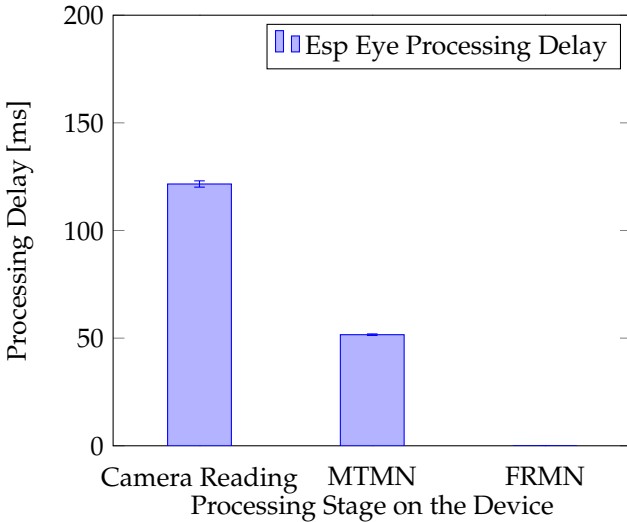

**Figure 7.** Esp Eye Processing Delay When no Face is Detected in Front of the Camera.

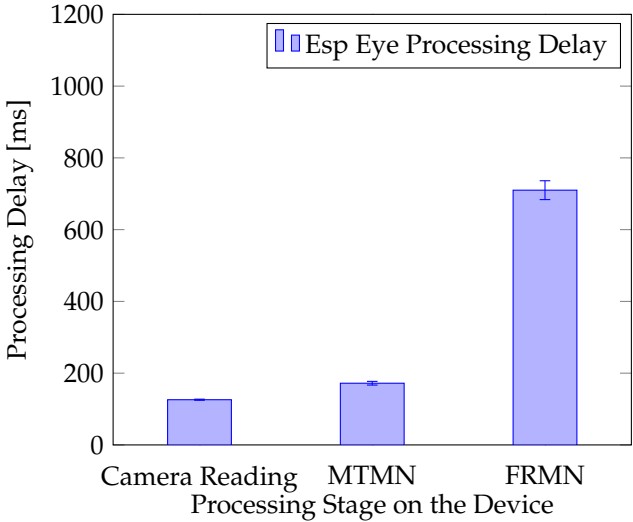

**Figure 8.** Esp Eye Processing Delay When a Face is Detected in Front of the Camera.

### 4.3. End-to-End Processing Delay

Many BC platforms suffer from a low TX rate. However, HLF executes TXs in an execute-and-order way. Thus, HLF gains significant speed in terms of processing delay to support large TX volumes per second. The end-to-end delay is measured by printing the timestamp at different individual processing steps to measure (*i*) image processing delay on the Esp Eye device, (*ii*) transmission delay of the collected image from Esp Eye to the IoT Gateway, (*iii*) preparation delay of the BC TX on the IoT Gateway, and (*iv*) BC TX submission delay from the IoT Gateway to the BC. The image is inserted in the ledger, and the BC service is finished when the BC TX reaches all peers. Figure 9 displays system processing stages starting with the Esp Eye image capturing until the image reaches all peers in the ledger, for an example TX. Similarly, as estimated earlier, face detection (i.e., MTMN) and recognition (i.e., FRMN) on Esp Eye take on average 1 s. However, if the face detection phase is not executed, the image processing is much shorter (i.e., at 200 ms). Sending the image from Esp Eye to the IoT Gateway takes almost 2 s. Furthermore, the time it takes from the IoT Gateway until the TX is submitted to HLF is very small compared to other functions. HLF consumes little more than 2 s. This is influenced by two configurable parameters, i.e., BatchTimeout and BatchSize. BatchTimeout is essential for this work, since it refers to a block-time limit. This work configures BatchTimeout at 2 s, which minimizes

the idle time of ordering peers. However, depending on the use case, those parameters may need to be tuned such that there is no bottleneck in TX arrivals. The end-to-end delay experienced in the system for this example BC TX is 5.3 s from the moment the image is taken until the inclusion of the BC TX in the BC. Therefore, almost real-time use cases can be supported using HLF as a communication backend.

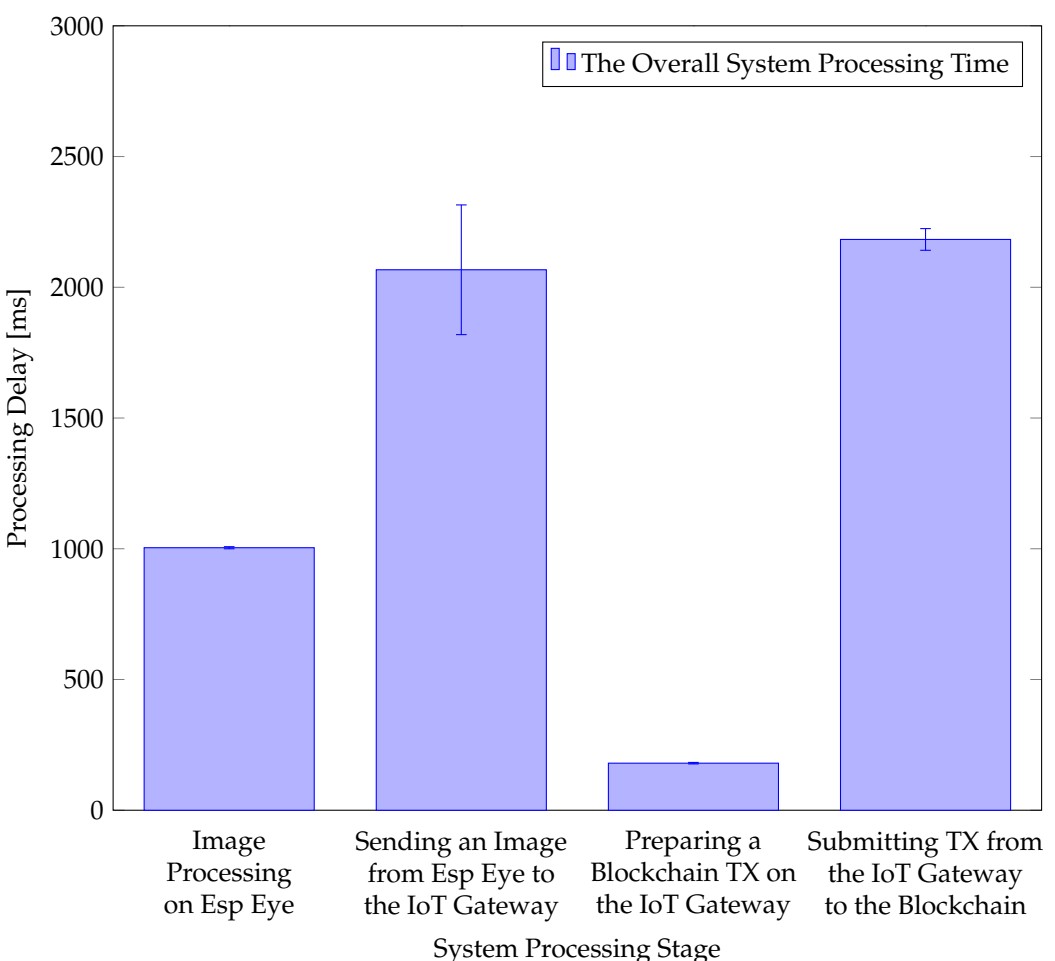

**Figure 9.** System Processing Delay.

*4.4. Energy Efficiency of Esp Eye*

The experimentation setup was used to measure the energy consumption of Esp Eye against image quality, face detection, and face recognition. After several tests with different image qualities and parameters, there was no difference experienced in Esp Eye energy consumption. Throughout the experiment, the energy consumption remained at the constant level of 600 mW (in total, the device consumed 600 mWh within an hour of operation), which allows for a 7-hour operation on an alkaline battery of 4200 mWh capacity. Furthermore, there is no difference in energy consumption when a face is recognized or no face is detected.

## 5. Summary and Future Work

This paper provides the first access management system that utilizes Artificial Intelligence (AI), Blockchains (BC), and Internet-of-Things (IoT) in an integrated use case. This use case is centered around access management without direct human intervention. Thus, the user needs to present their face in front of a camera to access a resource. The system takes an image of their face and checks whether this user has the right to access a given resource.

For that decision being taken, the face detection and recognition are performed directly on the IoT device. An MTCNN (Multi-Task Cascaded Convolutional Network) with MobileNets (MN) was deployed to detect faces. Furthermore, the Face Recognition model is based upon a Convolutional Neural Network (FRMN). To establish a good level of transparency, AI decisions on access rights and images taken by the sensor are stored in the immutable, tamper-resistant storage implemented with the help of the HyperLedger Fabric (HLF).

The system's performance was evaluated at an excellent level, where a 5.3 s end-to-end delay is reached. This value reads to be outstanding compared to well-established BC systems, such as Bitcoin (BTC) or Ethereum (ETH), having a block time configured at the level of 10 min and 10 s, respectively.

Additionally, the size of data to be persisted inside a block does not lead to high costs, as for a BTC or ETH case, since HLF does offer "unlimited" storage capacity due to its private, permissioned characteristic. At the same time, data privacy is ensured since the private ledger within the HLF implementation will only allow for authorized accesses by definition.

As this approach taken is considered to be a prototype, two current drawbacks will have to be taken care of in the next generation:

1. All face descriptors of recognized faces are hard-encoded on Esp Eye devices.
2. The HLF Transaction (TX) signing is not performed directly on Esp Eye devices, since it is delegated to the IoT Gateway device supporting this process on behalf of the sensor.

Apart from solving these aspects, future work will not only close the gaps as identified in the currently specified and prototyped system, but will further develop the IoT-BC integration into operational and reusable software functionality:

1. The face descriptor management will allow an IoT device to retrieve a remote directory of recognized faces.
2. Porting the HLF Software Development Kit (SDK) to IoT devices will enable an HLF TX submission directly from IoT devices.

**Author Contributions:** Conceptualization, E.S.; methodology, E.S.; software, E.E.; validation, E.S. and E.E.; formal analysis, E.S. and E.E.; investigation, E.S., E.E. and B.S.; resources, B.S.; data curation, E.S. and E.E.; writing—original draft preparation, E.E.; writing—review and editing, E.S., E.E. and B.S.; visualization, E.S. and E.E.; supervision, E.S. and B.S.; project administration, B.S.; funding acquisition, B.S. All authors have read and agreed to the published version of the manuscript.

**Funding:** This research was partially funded by *(a)* the University of Zürich and *(b)* the European Union's Horizon 2020 Research and Innovation Program under Grant Agreement No. 830927, the CONCORDIA project.

**Data Availability Statement:** The software generated in this work is publicly available on GitHub https://github.com/eesati/Master-Thesis/ (accessed on 25 January 2022).

**Conflicts of Interest:** The authors declare no conflict of interest.

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
