# Peer review of "IoT-Based Access Management Supported by AI and Blockchainsâ€"

_electronics, doi:10.3390/electronics11182971_

Round 1
Reviewer 1 Report
This paper combines new terms, such as AI, BC, and IoT, which is an interesting topic. However, a paper needs to have its own innovative content.
- The Introduction section should highlight the main contributions.
- The authors need to detail how to use AI and BC in Sections 3.2 and 3.3. The conclusion section mentioned that MT-CNN was deployed. However, we care about how to make use of it.
- It would be better to compare the experimental results with related work in the evaluation section.
- Figures 1, 2, 3,4, and 5 should be vector graphics.
Author Response
(i) We have expanded Sect. 1 to elaborate on the contributions of this paper.
(ii) We have expanded Sect. 2.2 and Sect. 3.3 to show how to use MTCNN on constrained devices.
(iii) We believe that our system is one of the first to perform AI, BC, and IoT efficiently. There are several ideas on how to compare our approach with other technologies. E.g., different blockchains can be used on the backend, or different AI frameworks can be used on IoT devices, etc. The systems will differ in precision, model sizes, computing efforts, reaction times, and the time to store a message in the blockchain. We will leave, however, those efforts for future work.
(iv) We will provide Fig 1-5 as vector graphics in the final version if/when the paper gets accepted.
Reviewer 2 Report
Summary: Authors have investigated some research gaps on integrated use of IoT, Blockchains, and AI. In their proposed research work, authors have clearly pointed that with AI lacking transparency in terms of providing predictors use of blockchain can complement to enable immutable track of AI decisions. Authors presented an interesting use case; an access management approach for combined use of AI, BC and IoT. Authors claimed that the existing research works on IoT, BC, and AI are only limited to research reviews and does provide any implementation in a use case.
Comments:
- Literature review is written well. In this section, authors have presented necessary details related to IoT, BC, AI and captured interesting information on the convergence of these technologies.
- Section 3.1- the sentence in the first paragraph starting with “every time an individual needs access………..” is incomplete.
- Novelty in this paper is not clear.
- Authors have clearly mentioned their approach is similar to the work presented in [23].
- Moreover, the results presented in the paper are similar to their previous work presented in [29]. What is different in this paper from [29]? It seems major contribution presented in this paper based on energy efficiency and end-to-end delay have already been published by the in [29].
- Algorithms used for face detection and recognition are not clearly explained.
Author Response
(i) We thank the reviewer for this comment.
(ii) We have fixed this deficit.
(iii) We have clarified in the Introduction and Sect. 3.2.2 that a specific BC is needed to store images directly in the BC, e.g., without the need for off-chain storage such as IPFS. We did not elaborate on those specific issues in [23,29].
(iv) We have improved the presentation of CNN-based MTMN and FRMN in Sect. 2 and Sect. 3.
Reviewer 3 Report
The authors present the work concerning the IoT-based access management supported by AI and Blockchains.
Here are my comments:
- Literature Review should be more elaborate as also there should be indicated the research gap and authors own contribution.
- Authors should add some remarks on the computational complexity of the proposed method compared with previous ones. The information about the computational, memory, and time complexity of the whole solution should be supplemented.
- The paper should include a chapter on research methodology, selected statistical tools, description of the data set, etc.
- The paper also lacks a detailed description of the implementation of the selected algorithm, diagrams, etc.
- It would be advisable to present diagrams of the main procedures of the implemented algorithms.
Author Response
(i) We have expanded the Introduction to show the authors' contributions.
(ii) We have evaluated the complexity of used algorithms cf. Sect. 2 and Sect. 3.
(iii) We didn't provide a methodology section, but the overall methodology was improved by providing additional information in Sect. 1 and Sect. 4.2.
(iv) New diagrams are provided in Sect. 2. We believe that everything is clear now.
(v) We evaluate a system developed by reusing different system components. Some supporting elements of the system are implemented, such as the infrastructure, smart contact, etc. All algorithms and their implementations are referenced in the paper.
Reviewer 4 Report
The paper does not present any novelty in the developments presented. There are many other works addressing the same problem and presenting similar solutions.
The methodology and verification of the investigation must be strengthened and improved.
The results are weak for a solution of this style.
Author Response
The paper describes our experience of building a real system composed of IoT, BC, and ML. We provide some interesting concepts, such as the choice of an appropriate blockchain, the smart contract implementation, the evaluation of the E2E, etc. We improved the methodology of the paper, cf. Sect 4. However, we do hold new results, as we will gather those in future work.
Round 2
Reviewer 3 Report
The explanations and improvements proposed by the authors seem to explain in detail the questions and comments from my first review. The revised version of the manuscript, in a better way, reflects the authors' contribution.
Author Response
We thank you for the time spent on this review and the positive comments we received in this review round.
Reviewer 4 Report
The paper shows an improvement at this point. The work shows better analysis and results.
Author Response

(The authors gave the same response as above.)
